

# Fingerprints of freeze-in dark matter
# in an early matter-dominated era

**Avik Banerjee[1]★ and Debtosh Chowdhury[2]†**

**1** Department of Physics, Chalmers University of Technology,
Fysikgården, 41296 Göteborg, Sweden
**2** Department of Physics, Indian Institute of Technology Kanpur,
Kanpur 208016, India

★ avik@chalmers.se † debtoshc@iitk.ac.in

## Abstract

We study the impact of an alternate cosmological history with an early matter-dominated epoch on the freeze-in production of dark matter. Such early matter domination is triggered by a meta-stable matter field dissipating into radiation. In general, the dissipation rate has a non-trivial temperature and scale factor dependence. Compared to the usual case of dark matter production via the freeze-in mechanism in a radiation-dominated universe, in this scenario, orders of magnitude larger coupling between the visible and the dark sector can be accommodated. Finally, as a proof of principle, we consider a specific model where the dark matter is produced by a sub-GeV dark photon having a kinetic mixing with the Standard Model photon. We point out that the parameter space of this model can be probed by the experiments in the presence of an early matter-dominated era.



# 1 Introduction

Despite the overwhelming evidences in favor of the 'dark matter' (DM) having a major share in the total energy budget of the Universe [1–4], barely anything is known about its nature and the origin. Most of the speculations regarding the particle identity of the DM revolve around the Weakly interacting massive particle (WIMP) framework, where the DM freezes-out from the thermal bath after a period of equilibrium with the visible matter. Despite strong theoretical motivations, dearth of any pointers from experiments [5–8] drive us to look for alternatives of WIMP paradigm. One of the notable alternative to the WIMP paradigm is the Feebly interacting massive particle (FIMP) framework [9, 10]. In this scenario, DM is produced out of thermal equilibrium via the so-called 'freeze-in' production mechanism. The interaction of the DM with the Standard Model (SM) particles in the freeze-in scenario needs to be extremely small such that they never attain a thermal equilibrium [11–29]. Such a tiny interaction strength makes freeze-in dark matter models challenging to detect at the experiments compared to the usual WIMP scenarios. For recent attempts to detect FIMP, see for example [30–33].

The present day relic abundance of DM produced by freeze-in mechanism significantly depends on the cosmological history as well as the initial DM abundance at the end of inflation. Any modification in the standard picture of the radiation domination after the inflation till the big bang nucleosynthesis (BBN) alters the production of the dark matter [34–55]. In this paper, we investigate the impact of a pre-BBN matter-dominated epoch on freeze-in DM production. A number of motivated scenarios beyond the Standard Model (BSM) of particle physics predict some meta-stable matter fields, for instance, string theory inspired moduli fields [56–58], supersymmetric condensates and gravitino [59, 60], scenarios with primordial black hole formation [61–64], inflaton fields [65, 66], curvaton [67], dilaton [68], Q-balls [69] etc. This kind of late-decaying species can give rise to an early matter-dominated (EMD) epoch after the inflation. Subsequently the decay of these long-lived particles into the radiation leads to a second phase of reheating at the end of which the universe once again enters into a radiation-dominated phase.

The temperature evolution of the universe depends on the detailed dynamics of how the matter field leading to the EMD dissipates into the radiation. In general, the dissipation rate depends on the instantaneous temperature of the thermal bath and the expansion rate of the universe [70]. In such a scenario, end of the EMD era is preceded by an epoch of entropy production which has a direct impact of diluting the present day relic abundance of the DM. Existing works illustrating such dilution of DM relic density generally consider a constant dissipation rate of the meta-stable field into the radiation [34–44]. Here instead, we consider a more general parametrization of the dissipation rate which depends on the temperature and the scale factor [70].

We show that the freeze-in production of DM is significantly affected by the presence of an EMD epoch with such generalized dissipation rate. In this scenario, orders of magnitude larger couplings between the DM and the visible matter are required for the freeze-in mechanism, compared to the usual case of freeze-in in a radiation-dominated universe. Moreover, we present some specific benchmark cases where the dissipation rates have non-trivial temperature and scale factor dependence and show that even larger couplings can be accommodated compared to an EMD epoch with constant dissipation rate. As a consequence, in this case even a freeze-in dark matter model can come under the microscope of experimental facilities, in stark contrast with freeze-in in a radiation-dominated universe.

As a proof of principle, we finally consider a motivated dark matter scenario where the freeze-in mechanism is facilitated through a dark photon portal having an extremely small kinetic mixing with the SM photon [31, 71–78]. In the absence of any matter-dominated epoch, the parameter space of this model is far beyond the reach of any current observations

or future experiments. We point out that for a sub-GeV dark photon, the kinetic mixing required to satisfy the observed relic density in the presence of an EMD epoch is accessible to the experiments. In particular, we show that the bounds from the supernova observations and beam-dump experiments can rule out a considerable part of the relevant parameter space.

The paper is organized as follows. In Sec. 2, we discuss the cosmological history with an early matter-dominated epoch. The impact of such matter-dominated epoch on the freeze-in DM production is shown in Sec. 3. In Sec. 4, we study the parameter space of a concrete model with a dark photon acting as a portal between the visible and the dark sector before concluding in Sec. 5.

## 2 Early matter domination with generalized dissipation

In this section, we explain the details of the alternate cosmological history with an early matter-dominated epoch. We consider that the total energy of the universe is dominated by a thermal plasma at the end of the inflation (characterized by a temperature $T_{\text{inf}}$)[1], leading to an early radiation domination (ERD). However, below some temperature $T_{\text{eq}}$ the total energy budget is assumed to be dominated by a new meta-stable species (denoted generically by $\phi$) red-shifting slower than the radiation[2]. This marks the beginning of an early matter-dominated epoch. The meta-stable field $\phi$ with equation of state $\omega \in (-1, 1/3)$ oscillates around its potential minima and would eventually decay into the SM radiation through its coupling to the SM particles. This would eventually lead to entropy production and a second phase of reheating towards a radiation-dominated (RD) universe below the temperature $T_{\text{RH}}$, compatible with the constraints from BBN [79–81]. We assume that the standard cosmological history prevails below $T_{\text{RH}}$.

The dissipation rate ($\Gamma_\phi$) of such a species $\phi$ mainly depends on the (i) temperature of the surrounding environment – in this case the temperature of the background radiation bath and (ii) on the expansion rate of the universe through its field dependent decay width to SM particles [70, 82–84]. For instance, it was shown that decay of scalar particle in a thermal environment leads to a temperature dependent decay width due to the back reaction of the plasma of decay products [85–92]. As an example, the dissipation rate of a moduli field in the early universe is given by $\Gamma_\phi \propto T^3$ for temperatures larger than its mass [88]. On the other hand, the dissipation rate of a meta-stable species with a potential $V(\phi) \propto \phi^p$ (which corresponds to $\omega = p - 2/p + 2$) can be written as

$$\Gamma_{\phi \to f\bar{f}} \propto m_\phi(t) \propto a^{-3(p-2)/(p+2)}, \text{ (for fermionic decay)},$$
$$\Gamma_{\phi \to \eta\eta} \propto m_\phi^{-1}(t) \propto a^{3(p-2)/(p+2)}, \text{ (for bosonic decay)},$$

where $m_\phi(t) \propto \langle\phi(t)\rangle^{(p-2)/2}$. We remain agnostic about the detailed dynamics of $\phi$ and express the envelop of $\phi$ averaged over larger time scales as $\langle\phi(t)\rangle \sim a^{-6/p+2}$ [93–95].

Motivated by these effects we parametrize the dissipation rate of the matter field into the radiation, as follows [70]

$$\Gamma_\phi = \hat{\Gamma}\left(\frac{T}{T_{\text{eq}}}\right)^n \left(\frac{a}{a_{\text{eq}}}\right)^k, \tag{1}$$

where $a_{\text{eq}}$ is the scale factor at $T_{\text{eq}}$ and $\hat{\Gamma}$ is a mass dimensional parameter dependent on the zero-temperature decay width[3]. The values of the exponents $n$ and $k$ depend on the detailed

---

[1]We assume that the post-inflationary reheating is instantaneous.

[2]In what follows we refer to $\phi$ as 'matter' because it is assumed to have a slower red-shift than the radiation.

[3]Note that we use different normalizations suitable for our context in Eq. (1) as compared to [54, 70].

Table 1: Representative cases illustrating the dependence of $\Gamma_\phi$ on the temperature and scale factor. The last column depicts its consequence on the temperature evolution during entropy production era.

| Dynamics of $\phi$ / $V(\phi)$ | $\Gamma_\phi$ | $(n, k, \omega)$ | $T(z)$ during $\text{EMD}_{\text{NA}}$ |
|---|---|---|---|
| $V(\phi) \sim \phi^2$ [70,96] | const., $m_\phi \gg T$ | $(0, 0, 0)$ | decreases with $z$ |
| | $T, m_\phi \ll T$ | $(1, 0, 0)$ | decreases with $z$ |
| $V(\phi) \sim \phi^p$ (fermionic decay of $\phi$) [82–84] | $\langle\phi\rangle^{\frac{p-2}{2}}, m_\phi \gg T$ | $(0, -\frac{3(p-2)}{p+2}, \frac{p-2}{p+2})$ | decreases with $z$ if $p \geq 1$ remains constant if $p = 1$ |
| $V(\phi) \sim \phi^p$ (bosonic decay of $\phi$) [82–84] | $\langle\phi\rangle^{-\frac{p-2}{2}}, m_\phi \gg T$ | $(0, \frac{3(p-2)}{p+2}, \frac{p-2}{p+2})$ | decreases with $z$ if $p \geq 0$ |
| Rotating scalar, $V(\phi) \sim \phi^\dagger\phi$ | $\langle\phi\rangle^{-2}, m_\phi \gg T$ | $(0, 3, 0)$ | increases with $z$ |
| $\log\left(\frac{\phi}{T}\right) F_{\mu\nu} F^{\mu\nu}$ interaction [70,89,90] | $\frac{T^3}{\langle\phi\rangle^2}, m_\phi \ll T$ | $(3, 3, 0)$ | increases with $z$ |
| Oscillating scalar, $V(\phi) \sim \phi^p$ | $\langle\phi\rangle^{-1}, m_\phi \gg T$ | $(0, \frac{6}{p+2}, \frac{p-2}{p+2})$ | remains constant if $p = 2$ decreases with $z$ if $p \geq 2$ |
| $\log\left(\frac{\phi}{T}\right) F_{\mu\nu} F^{\mu\nu}$ interaction [70,89,90] | $\frac{T^2}{\langle\phi\rangle}, m_\phi \ll T$ | $(2, \frac{6}{p+2}, \frac{p-2}{p+2})$ | |

dynamics of the oscillating matter field and its interactions with the thermal plasma. Different possibilities for $n$ and $k$ in specific scenarios are discussed in [54,70,89,90]. In Tab. 1, we list some representative cases where the dissipation rate depends on the temperature of the thermal bath and $\langle\phi(t)\rangle$. The evolution of energy densities of the late-decaying matter $\rho_\phi$ and radiation $\rho_\gamma$ are given by the following equations

$$\dot{\rho}_\phi + 3(1+\omega)H\rho_\phi = -(1+\omega)\Gamma_\phi\rho_\phi \,, \tag{2}$$

$$\dot{\rho}_\gamma + 4H\rho_\gamma = (1+\omega)\Gamma_\phi\rho_\phi \,, \tag{3}$$

where the Hubble expansion rate is $H = \sqrt{\rho_\phi + \rho_\gamma}/\sqrt{3}M_p$ with $M_p$ being the reduced Planck mass. In order to solve Eq. (3), we make the crucial assumption that the radiation produced from $\phi$ thermalizes instantaneously. This allows us to relate $\rho_\gamma$ with an instantaneous bath temperature as $\rho_\gamma = (\pi^2/30)g_*(T) T^4$. For simplicity we assume that the effective energetic and entropic degrees of freedom are constant, $g_*(T) = g_{*S}(T) = 106.75$. It is convenient to express the solutions for the energy densities in terms of a dimensionless variable $z \equiv a/a_{\text{eq}}$ (with $dz = zHdt$).

In Fig. 1 we present the variation of the temperature of radiation bath as a function of $z$ for different values of $(n, k, w)$, keeping $T_{\text{eq}}$ and $T_{\text{RH}}$ fixed. For $0 \leq n < 4$ and $\delta \equiv 5 - 2n + 2k - 3\omega > 0$, the EMD epoch can be further divided into two regions: an epoch of adiabatic evolution of the temperature ($\text{EMD}_{\text{A}}$), followed by an entropy production phase where the thermal plasma becomes non-adiabatic ($\text{EMD}_{\text{NA}}$). The approximate solutions for the Eq. (2) and Eq. (3) in the EMD epoch are given by

$$\rho_\phi(z) \simeq \rho_\gamma(T_{\text{eq}}) z^{-3(1+\omega)} \,,$$

$$\rho_\gamma(z) = z^{-4} \left[ \rho_\gamma(T_{\text{eq}})^{\frac{4-n}{4}} + \frac{\sqrt{3}M_p \hat{\Gamma}(4-n)(1+\omega)}{2\delta \rho_\gamma(T_{\text{eq}})^{\frac{(n-2)}{4}}} \left( z^{\delta/2} - 1 \right) \right]^{\frac{4}{4-n}} \,. \tag{4}$$

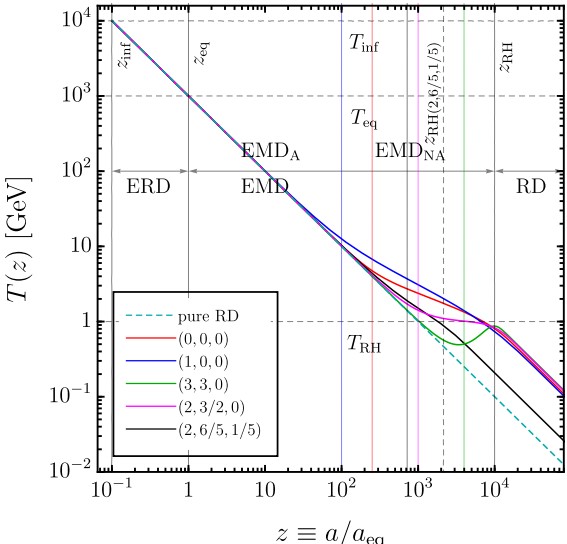

Figure 1: Evolution of the thermal plasma temperature as a function of $z$, obtained by numerically solving Eq. (2) and Eq. (3) for different values of $(n, k, \omega)$ as given in the legend. The colored vertical lines show the approximate values of $z_{\text{NA}}$ for different cases with specific $(n, k, \omega)$. For the purpose of illustration, we assume $T_{\text{inf}} = 10^4$ GeV, $T_{\text{eq}} = 10^3$ GeV and, $T_{\text{RH}} = 1$ GeV. Note that the value of $z_{\text{RH}}$ depends on $\omega$. For $(2,6/5,1/5)$ case, the value of $z_{\text{RH}}$ is shown by the vertical dashed line.

Table 2: Approximate expressions for the evolution of $T(z)$ and $H(z)$ in different cosmological epochs.

| Epoch | $z$ | $T(z)$ | $H(z)$ |
|---|---|---|---|
| ERD | $z_{\text{inf}} < z < 1$ | $\frac{T_{\text{eq}}}{z}$ | $\sqrt{\frac{\rho_\gamma(T_{\text{eq}})}{3M_p^2}} z^{-2}$ |
| EMD$_A$ | $1 < z < z_{\text{NA}}$ | $\frac{T_{\text{eq}}}{z}$ | $\sqrt{\frac{\rho_\gamma(T_{\text{eq}})}{3M_p^2}} z^{-\frac{3}{2}(1+\omega)}$ |
| EMD$_{\text{NA}}$ | $z_{\text{NA}} < z < z_{\text{RH}}$ | $T_{\text{RH}} \left(\frac{z}{z_{\text{RH}}}\right)^{\frac{\delta-8+2n}{8-2n}}$ | $\sqrt{\frac{\rho_\gamma(T_{\text{eq}})}{3M_p^2}} z^{-\frac{3}{2}(1+\omega)}$ |
| RD | $z_{\text{RH}} < z$ | $T_{\text{eq}} z_{\text{RH}}^{\frac{1-3\omega}{4}} z^{-1}$ | $\frac{\sqrt{\rho_\gamma(T_{\text{RH}})}}{\sqrt{3}M_p} \left(\frac{z}{z_{\text{RH}}}\right)^{-2}$ |

For the $\omega = 0$ case, our generic results conform with those obtained in [70]. Approximate analytic expressions for the variation of $T(z)$ and $H(z)$ in the various epochs are shown in Tab. 2. The value of $z$ at the beginning of the EMD$_{\text{NA}}$ can be calculated as

$$z_{\text{NA}} = \left[ 1 + \frac{2\delta\rho_\gamma(T_{\text{eq}})^{\frac{1}{2}}}{\sqrt{3}M_p\hat{\Gamma}(4-n)(1+\omega)} \right]^{2/\delta}. \tag{5}$$

The temperature evolution during EMD$_{\text{NA}}$ era is sensitive to the exact temperature and scale factor dependence in Eq. (1). For $n \geq 4$ or $\delta < 0$ the adiabatic evolution ends by an instantaneous reheating at $T_{\text{RH}}$ which we do not consider here. In Fig. 1, different cases of $(n, k, \omega)$ lead to qualitatively different evolution of the temperature during the EMD$_{\text{NA}}$ epoch. During

the $\text{EMD}_{\text{NA}}$ era, the temperature remains constant for $2k = 3(1 + \omega)$ (the $(2, 3/2, 0)$ case in Fig. 1), while $T(z)$ grows with $z$ for $2k > 3(1 + \omega)$ (the $(3, 3, 0)$ case in Fig. 1). End of the EMD era is characterized by the condition $\rho_\gamma(T_{\text{RH}}) = \rho_\phi(T_{\text{RH}})$, and the corresponding $z$ is given as

$$z_{\text{RH}} = \left[ \frac{2\delta\rho_\gamma(T_{\text{eq}})^{\frac{1}{2}}}{\sqrt{3}M_p\hat{\Gamma}(4-n)(1+\omega)} \right]^{\frac{4}{2\delta-(4-n)(1-3\omega)}} = \left( \frac{T_{\text{RH}}}{T_{\text{eq}}} \right)^{-\frac{4}{3(1+\omega)}}. \tag{6}$$

The scale factor at today, expressed in terms of $z$ is given by $z_0 = (T_{\text{eq}}/T_0)(T_{\text{RH}}/T_{\text{eq}})^{\frac{(3\omega-1)}{3(1+\omega)}}$. Note that for $\omega \in (-1, 1/3)$, $z_0$ is always greater than $(T_{\text{eq}}/T_0)$ which would have been the value of $z_0$ in the absence of matter domination.

# 3 Dark matter freeze-in during matter domination

In this section we discuss the impact of an EMD epoch with generalized dissipation of $\phi$ on the freeze-in production of dark matter. Simply speaking, the entropy production during the non-adiabatic phase of the EMD era induces a dilution in the relic abundance of the dark matter. However, $T$ and $z$ dependent dissipation rate of the meta-stable matter changes the amount of dilution of the DM relic. Thus, a larger production rate for the DM may be required to reproduce the observed abundance at the present time.

To illustrate our point, we consider a simple setup where the DM ($\chi$) is produced in pair by the annihilation of two SM particles via the process $\text{SM} + \text{SM} \rightarrow X^* \rightarrow \chi + \chi$. Here $X$ acts as a portal between the dark sector and the visible sector which can, for instance, be a dark photon, a $Z'$, or a RH neutrino with extremely small couplings with the SM. The small coupling with the SM is necessary to ensure that $X$ is never in the thermal bath. Coupling between the $X$ and the DM can, however, be as large as $\mathcal{O}(1)$.

The Boltzmann equation for the evolution of dark matter number density ($n_\chi$) is given by

$$\dot{n_\chi} + 3Hn_\chi = R(T), \tag{7}$$

where $R(T)$ denotes the freeze-in production rate of the dark matter. The rate for the $2 \rightarrow 2$ scattering $\text{SM} + \text{SM} \rightarrow X^* \rightarrow \chi + \chi$ can be written in its full glory as [77]

$$R(T) = \frac{T}{2048\pi^6} \int ds\sqrt{s}K_1\left(\frac{\sqrt{s}}{T}\right)\sqrt{1 - \frac{4m_\chi^2}{s}}\sqrt{1 - \frac{4m_B^2}{s}} \int d\Omega\,|\mathcal{M}|^2, \tag{8}$$

where $m_B$ generically represents the mass of a SM particle in the thermal bath and $|\mathcal{M}|^2$ is the amplitude square summed over initial and final states. In Eq. (8), we assume that both the SM particles and the dark matter follow Maxwell-Boltzmann distribution and the interactions between them are CP invariant. The production rate given above can be parametrized in a simple form depending on the hierarchy between the DM mass ($m_\chi$), the mediator mass ($M_X$) and the masses of the bath particles ($m_B$) as

$$R(T) \simeq \begin{cases} \lambda_1^2 T^p e^{-\frac{2m_B}{T}}, & m_B \gg M_X, \quad M_X \gg 2m_\chi, \\[2mm] \lambda_1^2 T^p, & m_B \ll M_X \ll T, \quad M_X \gg 2m_\chi, \\[2mm] \lambda_2^2 M_X^p\left(\frac{\pi T}{\Gamma_X}\right)K_1\left(\frac{M_X}{T}\right), & m_B \ll M_X \sim T, \quad M_X \gg 2m_\chi, \\[2mm] \lambda_3^2 \frac{T^{p+4}}{M_X^4} e^{-\frac{2m_B}{T}}, & m_B, T \ll M_X, \quad M_X \gg 2m_\chi, \\[2mm] \lambda_1^2 T^p e^{-\frac{2}{T}\max\{m_\chi, m_B\}}, & M_X \ll 2m_\chi. \end{cases} \tag{9}$$

Here the exponent $p = 4, 6, 8, ...$ depends on the dimension of the operators describing the interactions of $X$ with the SM and the DM. For example, $p = 4$ arises for renormalizable interactions while $p > 4$ comes from higher dimensional operators [97]. The coefficients $\lambda_{1,2,3}^2$, with dimension $[M]^{4-p}$, vary depending on the details of specific model as well as relative values of $M_X$ and $T$.

We define a comoving number density of the DM as $Y_\chi(z) \equiv n_\chi z^3$ and express the Boltzmann equation in terms of the dimensionless parameter $z$ as

$$\frac{dY_\chi(z)}{dz} = \frac{z^2 R(T(z))}{H(z)} \,. \tag{10}$$

The present day relic density of dark matter is given by

$$\Omega_\chi h^2 = \frac{m_\chi n_\chi(T_0)}{\rho_c} = \frac{m_\chi Y_\chi(z_0)}{\rho_c z_0^3} \,, \tag{11}$$

where $\rho_c$ denotes the critical density today and

$$Y_\chi(z_0) = Y_\chi(z_{\text{inf}}) + \int_{z_{\text{inf}}}^{z_0} dz \frac{z^2 R(T(z))}{H(z)} \,. \tag{12}$$

Note that, during the EMD era $z$ dependence of $H(z)$ and $T(z)$ are different compared to radiation-dominated epoch, thereby also affecting the rate $R(T)$. For the freeze-in DM production, we assume that the initial DM abundance at the end of the inflation $Y_\chi(z_{\text{inf}}) = 0$. The effect of entropy injection during the $\text{EMD}_{\text{NA}}$ era is encoded into the $z_0^3 > (T_{\text{eq}}/T_0)^3$ factor, as given in Eq. (11). Thus, in the presence of an EMD epoch larger value of $Y_\chi(z_0)$ is needed to reproduce the observed $\Omega_\chi h^2 = 0.12$, compared to the requirement in a purely radiation-dominated universe. This in turn implies that even for the freeze-in production larger coupling between the dark sector and the visible sector can be accommodated in the presence of EMD era.

The impact of matter domination on $Y_\chi(z)$ depends on the temperature dependence of the DM production rate. For $R(T) \propto T^p \exp(-2m_\chi/T)$ with $p > 4$, majority of the DM is produced at the ERD regime at higher temperatures, giving rise to the UV freeze-in. In this case, $Y_\chi(z)$ is not sensitive to the variation of $(n, k, \omega)$ during the EMD epoch since the DM number density already saturates at much smaller $z$. Here we mention en passant that UV freeze-in of DM can be boosted during an inflaton dominated epoch if the inflaton decays into the radiation with a generalized rate as given in Eq. (1) [54]. On the other hand, IR freeze-in dominates for $p = 4$, and majority of the DM production may happen during the EMD epoch depending on the DM mass and the values of $(n, k, \omega)$. In both scenarios, however, the entropy production during the final stage of matter domination dilutes the relic of the DM produced.

For the sake of presentation, we assume $p = 4$, $m_\chi = 1$ GeV, $M_X = 10$ GeV, and $\lambda_1 = 2 \times 10^{-14} = 2\lambda_2 = \lambda_3/8\sqrt{3}$[4]. In the left panel of Fig. 2, we show the evolution of $Y_\chi(z)$ for a fixed $T_{\text{RH}} = 1$ GeV. The horizontal dashed lines represent the contours satisfying $\Omega_\chi h^2 = 0.12$ for different cases. We choose the parameters in such a way that the DM yield approximately saturates the $\Omega_\chi h^2 = 0.12$ limit in a purely radiation dominated universe (dashed cyan contours). In contrast, the DM is always under-abundant for the other scenarios in presence of an EMD epoch. The ratio of the relic densities of the DM in the presence of an EMD era to that in a purely radiation dominated universe is given by

$$\frac{\Omega_\chi h^2}{\Omega_\chi h_{\text{RD}}^2} = \frac{Y_\chi(z_0)}{Y_\chi^{\text{RD}}(z_0)} \left(\frac{z_0^{\text{RD}}}{z_0}\right)^3 = \frac{Y_\chi(z_0)}{Y_\chi^{\text{RD}}(z_0)} \left(\frac{T_{\text{RH}}}{T_{\text{eq}}}\right)^{\frac{1-3\omega}{1+\omega}} \,, \tag{13}$$

---

[4]The ratios $\lambda_1^2/\lambda_2^2$ and $\lambda_1^2/\lambda_3^2$ can be fixed by performing the integral over $s$ in Eq. (8) for a fixed value of $p(= 4)$.

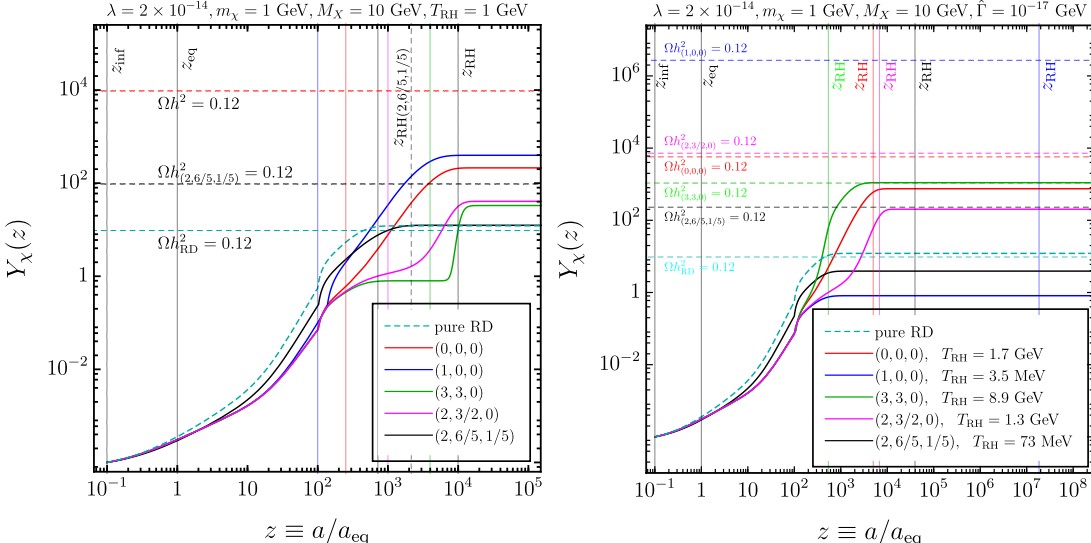

Figure 2: In the left panel, we show $Y_\chi(z)$ as a function of $z$ keeping a fixed value of $T_{\text{RH}} = 1$ GeV. The colored vertical lines denote the position of $z_{\text{NA}}$ for different cases. The horizontal dashed lines present the contours of $\Omega_\chi h^2 = 0.12$. In the right panel, we present the variation of $Y_\chi(z)$ keeping $\hat{\Gamma}$ fixed. In this case, the colored vertical lines represent the corresponding values of $z_{\text{RH}}$. In both panels we fix $T_{\text{inf}} = 10^4$ GeV and $T_{\text{eq}} = 10^3$ GeV.

where the factor $\left(\frac{T_{\text{RH}}}{T_{\text{eq}}}\right)^{\frac{1-3\omega}{1+\omega}}$ encodes the entropy dilution. For the benchmark parameter choice used in the left panel of Fig. 2, this ratio varies between $\sim [0.1-0.001]$. As a consequence, it is evident that one would need a larger coupling between the DM and the SM particles to achieve the observed relic abundance in presence of matter domination. Interestingly, for $(3,3,0)$ and $(2,3/2,0)$, a second stage of DM production occurs, since the temperature starts to increase during the $\text{EMD}_{\text{NA}}$ era.

In the right panel of Fig. 2, we present the variation of $Y_\chi(z)$ keeping $\hat{\Gamma} = 10^{-17}$ GeV fixed. Different values of $(n, k, \omega)$ thus lead to different values of $T_{\text{RH}}$ ranging from 3.5 MeV to a few GeV (corresponding values of $z_{\text{RH}}$ are shown by colored vertical lines). Since the DM production ceases below $T \sim m_\chi$, the saturation of $Y_\chi(z)$ does not necessarily coincide with $T = T_{\text{RH}}$. For example, for the blue line in Fig. 2 corresponding to $(1,0,0)$, $Y_\chi(z)$ is saturated much before the end of EMD era. In this case, the dilution due to entropy production during the matter domination is maximum, thereby yielding an under-abundance of DM at present day. In contrast, for $(0,0,0)$ (red line), the DM number density is saturated just after the end of EMD, causing less dilution compared to $(1,0,0)$.

# 4 Case study: dark photon portal to dark matter

Finally, we focus on a specific model where a fermionic dark matter is produced by the freeze-in mechanism via a dark photon portal interaction. In the presence of an EMD epoch with generalized dissipation of $\phi$, this model comes under the scanner of terrestrial and astrophysical experiments.

We consider a Dirac fermion dark matter $\chi$, charged under a dark gauged $U(1)_D$ with a dark photon ($A'_\mu$) which has a kinetic mixing with the SM photon. The relevant part of the

Lagrangian involving $A'_\mu$ and $\chi$ is given by

$$\mathcal{L} = -\frac{1}{4}F'_{\mu\nu}F'^{\mu\nu} - \frac{\epsilon}{2}F'_{\mu\nu}F^{\mu\nu} + \frac{1}{2}M_{A'}^2 A'_\mu A'^\mu + \bar{\chi}\left(i\slashed{\partial} - m_\chi\right)\chi + g_D\bar{\chi}\gamma^\mu A'_\mu \chi, \qquad (14)$$

where $\epsilon$ and $g_D \equiv \sqrt{4\pi\alpha_D}$ denote the kinetic mixing parameter and the dark gauge coupling, respectively.[5] After diagonalizing the kinetic mixing term, the dark photon interacts with the SM particles with an interaction strength proportional to $\epsilon Q$, where $Q$ is the electric charge of the corresponding SM field. We are primarily interested in the mass range $M_{A'} \lesssim 1$ GeV, where the DM production via freeze-in is dominated by the process $f\bar{f} \to A' \to \chi\bar{\chi}$ ($f$ being the SM fermions). The production rate for this process is given by [77]

$$R(T) = \frac{N_c Q_f^2 \alpha_e \alpha_D \epsilon^2 T}{6\pi^3} \int ds \sqrt{\left(1 - \frac{4m_\chi^2}{s}\right)\left(1 - \frac{4m_f^2}{s}\right)} \frac{s^{5/2}\left(1 + \frac{2m_\chi^2}{s}\right)\left(1 + \frac{2m_f^2}{s}\right)}{(s - M_{A'}^2)^2 + M_{A'}^2\Gamma_{A'}^2} K_1\left(\frac{\sqrt{s}}{T}\right), \tag{15}$$

where $N_c$ and $Q_f$ denote the number of colors and electric charge of the initial state fermions respectively. The decay width of the dark photon $\Gamma_{A'}$, dominated by its decay into a pair of DM when $M_{A'} > 2m_\chi$, is given as [77]

$$\Gamma_{A'} \simeq \frac{\alpha_D M_{A'}}{3}\left(1 + \frac{2m_\chi^2}{M_{A'}^2}\right)\sqrt{1 - \frac{4m_\chi^2}{M_{A'}^2}}. \tag{16}$$

The rate given in Eq. (15) can be approximated depending on the hierarchy between $M_{A'}$ and $m_\chi$, as shown in Eq. (9). Two different scenarios are of interest to us. In the first case, $M_{A'} \gg 2m_\chi$ so that the $A'_\mu$ can be produced on-shell by $s$-channel resonance and subsequently decay into a pair of DM particles. In the left panel of Fig. 3, we present the contours satisfying $\Omega_\chi h^2 = 0.12$ in the $\epsilon - M_{A'}$ plane with specific values of $(n, k, \omega)$. For the purpose of illustration, we assume $M_{A'} = 10m_\chi$ and $\alpha_D = 10^{-3}$, as well as $T_{\text{inf}} = 10^5$ GeV, $T_{\text{eq}} = 10^4$ GeV, and $T_{\text{RH}} = 8$ MeV. We chose the sub-GeV mass range for $M_{A'}$ and $m_\chi$ to access the sweet-spot where the effect of EMD epoch can push the freeze-in parameter space within the reach of experimental observations.

In this range of parameters, the dark photons produced by the inverse decay processes $\text{SM} + \text{SM} \to A'_\mu$ rapidly decays into dark matter with the rate given by Eq. (15), since it is the dominant decay channel for $A'_\mu$. As a result such inverse decays do not lead to a thermal abundance of $A'_\mu$. Thus the condition $R(T) \ll n_f^{\text{eq}}H$ (where $n_f^{\text{eq}}$ is the equilibrium number density of the SM fermions) is sufficient to guarantee that both the dark matter and the dark photon are not in thermal equilibrium with the visible sector. The grey shaded area in Fig. 3 shows the region where the freeze-in condition $R(T) \ll n_f^{\text{eq}}H$ is violated. Although, the freeze-in condition depends on the values of $(n, k, \omega)$, we only show the most restrictive condition. We also overlap the constraints from the supernova cooling [98] on the dark photon mass and the kinetic mixing. Note that for $(n, k, \omega) = (3, 3, 0)$ and $(2, 3/2, 0)$, some part of the parameter space is ruled out by the supernova observations. In the above plot we have set $g_*(T) = 10$. An actual variation of $g_*(T)$ with respect to temperature does not change the overall picture presented in this work. For $M_{A'}, m_\chi < T_{\text{RH}}$, majority of the DM is produced after the end of the matter-dominated epoch since the freeze-in production is dominated by the IR contributions.

---

[5]Above the weak scale the $U(1)_D$ gauge boson has a kinetic mixing with the hypercharge $B_\mu$ which eventually gives rise to the kinetic mixing term between the $F'_{\mu\nu}$ and $F_{\mu\nu}$ below the electroweak symmetry breaking scale. In addition to that, a mixing between $A'_\mu$ and the $Z_\mu$ boson is also induced. However, for the dark matter phenomenology discussed in this section with $M_{A'} \lesssim 1$ GeV, the effect of the $A'_\mu - Z_\mu$ mixing is negligible.

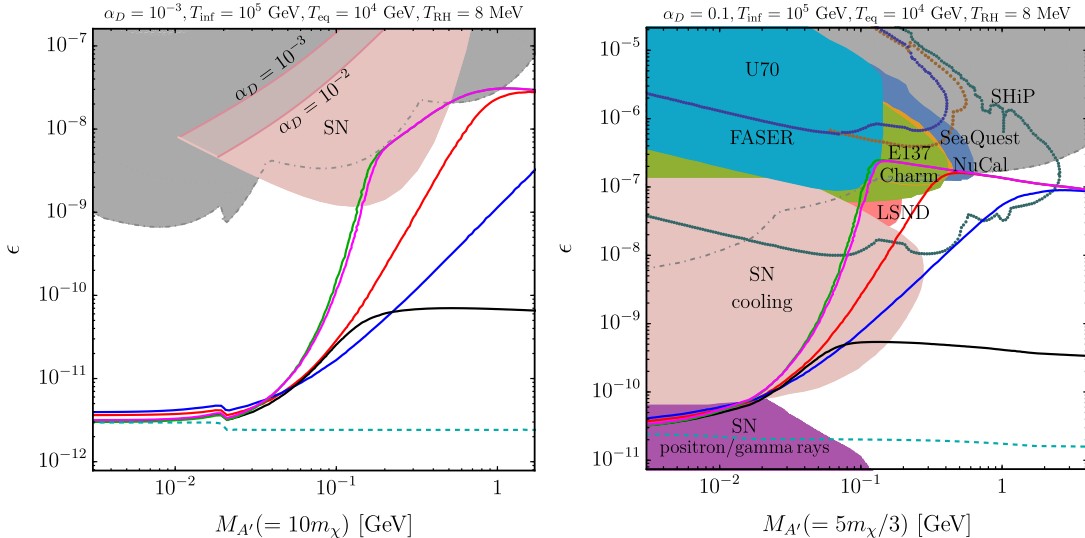

Figure 3: The contours satisfying $\Omega_\chi h^2 = 0.12$ are shown for $M_{A'} = 10m_\chi$ (left panel) and $M_{A'} = 5m_\chi/3$ (right panel). The dark grey region is not accessible for the freeze-in production as the condition $R(T)/n_f^{\rm eq}H \ll 1$ is violated and as a consequence DM comes in thermal equilibrium with the visible matter. The color coding for different $\Omega_\chi h^2$ contours are same as in Fig. 2. In the left panel, we overlap the supernova cooling constraints [98] in the presence of non-zero branching ratios for the decay of dark photon into dark matter. On the other hand, in right panel the existing constraints from different beam dump experiments [99–104], supernova bounds [98, 105, 106] and the projected bounds for future experiments like SHiP, FASER, and SeaQuest [107–110] are superimposed assuming the dark photon decays into SM particles only.

As a consequence, all the contours satisfying $\Omega_\chi h^2 = 0.12$ roughly overlap with the RD case in that region of the parameter space.

When $M_{A'} < 2m_\chi$, $A'_\mu$ decay to dark matter is kinematically suppressed. For an ultralight dark photon with $M_{A'} \ll m_\chi$, the DM behaves as a millicharged particle which has been discussed in [30, 111, 112]. Here instead, we focus on the mass range $m_\chi < M_{A'} < 2m_\chi$, where $f\bar{f} \to A'^* \to \chi\bar{\chi}$ still dominates the DM production. In this case, we plot in the right panel of Fig. 3, the contours satisfying $\Omega_\chi h^2 = 0.12$ assuming $M_{A'} = 5m_\chi/3$, $\alpha_D = 0.1$ and $T_{\rm RH} = 8$ MeV. The constraints on the kinetic mixing parameter from supernova cooling [98,105,106] as well as several beam dump experiments [99–104] rule out a significant region of the parameter space with sub-GeV masses. Moreover, future experiments like SHiP [107, 108, 110] will also probe a large part of the parameter space untouched by the current generation experiments. Similar to the left panel of Fig. 3, the grey shaded area in the right panel shows the region where the freeze-in condition, $R(T)/n_f^{\rm eq}H \ll 1$, for the DM production via $f\bar{f} \to A'^* \to \chi\bar{\chi}$ process is violated. However, in this case we need to consider additional constraint on the kinetic mixing parameter to ensure that the dark photon, produced via inverse decay processes SM + SM $\to A'_\mu$, can not achieve thermal equilibrium. To this end, we compare the inverse decay rate $R_{\rm inv}(T)$ to the Hubble expansion rate ($R_{\rm inv}(T) \ll n_{\rm SM}^{\rm eq}H$). As a consequence, one can safely neglect other DM production channels such as $A'_\mu + A'_\nu \to \chi + \bar{\chi}$. We observe that this condition does not rule out any additional parameter space which was untouched by the grey area or the existing observations.

# 5 Conclusions

The main message of this work is that the freeze-in production of the dark matter is significantly altered in the presence of an early matter-dominated epoch prior to the BBN. Many motivated BSM models predict late-decaying fields that can lead to a pre-BBN matter-dominated epoch. The transition from the matter domination to the usual radiation domination occurs through the dissipation of the matter field into radiation. The generic dependence of the dissipation rate on the instantaneous temperature of the thermal bath and the expansion rate of the universe leads to a qualitatively different evolution of the temperature and entropy of the thermal plasma.

These alterations of the background cosmology impacts the dark matter production rate in the early universe. Due to an additional entropy injection during the matter-dominated epoch, a larger coupling between the visible and the dark sector is essential to produce the observed relic density of dark matter via the freeze-in mechanism. This leads to the interesting possibility of probing the freeze-in dark matter models in experiments, which would otherwise be impossible in the usual radiation-dominated cosmology.

We would like to point out that majority of the analysis presented in this work is model independent. We have elucidated the DM production via freeze-in for a number of well-motivated cases with different temperature and scale factor dependence of the dissipation rate. Indeed, our study quantitatively shows that orders of magnitude larger couplings are required to produce the correct freeze-in relic abundance in the presence of early matter domination.

Finally we concentrate on a concrete model consisting a fermionic dark matter interacting with the SM via a dark photon portal to illustrate this phenomenon. We show that the relevant parameter space for freeze-in DM production with a sub-GeV dark photon mediator comes under the scanner of supernova observations and other terrestrial experiments.

## Acknowledgments

The authors thank Marcos A. G. Garcia, Andrew J. Long, and Yann Mambrini for valuable comments on the manuscript. A.B. acknowledges support from the Knut and Alice Wallenberg foundation (Grant KAW 2017.0100, SHIFT project). This research of D.C. is supported by an initiation grant IITK/PHY/2019413 at IIT Kanpur and by a DST-SERB grant SERB/CRG/2021/007579.

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
