# Peer review of "Fingerprints of freeze-in dark matter in an early matter-dominated era"

_SciPost Physics, doi:SciPost Phys. 13, 022 (2022)_

## Round 2 · Referee Report · Anonymous (Referee 1) · 2022-6-1

Report

This paper considers the freeze-in production of dark matter during an early matter-dominated (EMD) epoch. A common assumption in these scenarios is that the field responsible for EMD decays with a constant decay rate into SM particles (as would be the case for a particle with mass much larger than the temperature). The authors consider a complementary range of models where the decay rate instead is scale-factor and/or temperature dependent. This can occur if the mass of the decaying particle is comparable to the temperature (such that the backreaction from the final state particles is significant) or if the potential of the decaying particle is not quadratic (in models where this field is an oscillating scalar). This is an interesting and important exercise since it can parametrically change our expectations for the relationship between the DM mass and coupling to SM particles. As the authors point out, in some models this leads to significant improvements in the testability of freeze-in models, which are notoriusly difficult to probe due to the (usually) tiny couplings needed to saturate the relic abundance.

The paper is well written and mostly clear. I will recommend the preprint for publication after the authors have addressed the following minor issues: 1) In the first paragraph the authors claim that "Such a tiny interaction strength renders freeze-in dark matter invisible to the experiments". This is not true; in fact, freeze-in DM is one of the main benchmarks targeted by low mass direct detection experiments. See, for example, Fig. 21 in https://arxiv.org/pdf/1904.07915.pdf and the references in the figure caption.

2) The detectability of freeze-in in an EMD cosmology was also mentioned in https://arxiv.org/pdf/1807.01730.pdf in Sec. III.F in a dark photon model similar to the one considered by the authors (in much more generality).

3) While the authors provide an extensive set of references that describe various models for the dissipation behaviour of the EMD field, I think the paper could benefit from a more explicit mapping between benchmark models considered in Table I and these references or concrete physical scenarios. Specifically, I think Table I can be extended with one or two extra columns giving a reference where the parameters in column two appear and a brief description of the scalar potential or particle mass that gives rise to this behaviour (if it possible to describe them briefly).

4) The benchmark models in Table I and Fig. 1, as well as several equations are close to those of https://arxiv.org/pdf/2007.04328.pdf. While the authors do cite this paper, they should make the citation more prominent whenever their results are used or used as direct motivation for the authors' study; if the authors have reproduced analytic results from that work, they should comment on whether they find agreement or not.

5) On page 7, the authors say "it is evident that one would need a larger coupling between the DM and the SM particles to achieve the observed relic abundance in presence of matter domination" after saying that their chosen lambda gives rise to overabundance. This seems countintuitive since the production rate in Eq. 3.3 (and indeed the freeze-in abundance) is proportional to the coupling; smaller abundance should therefore correspond to smaller couplings, as typical for freeze-in. Please clarify this point.

6) In Fig. 3 the authors show a region in gray in which thermal equilibrium is attained and therefore freeze-in is not possible. From what I can tell, this comes from comparing the 2->2 rate SM SM -> chi chi with the Hubble rate. However there may be other processes that are important, such as inverse decays SM SM -> A', and the A' thermalizing amongst themselves and chi via U(1)_D gauge interactions. The authors should comment on whether these are parametrically similar to the constraint shown.

7) In Fig. 3 there may be additional relevant bounds coming from supernova, see https://arxiv.org/pdf/1901.08596.pdf (visibly decaying dark photons) and https://arxiv.org/pdf/1905.09284.pdf (dark photons decaying to dark matter).

---

## Round 3 · Referee Report · Anonymous (Referee 1) · 2022-7-13

Report

The authors have addressed my comments and implemented my suggestions. I therefore recommend this paper for publication.

---

## Round 3 · Author Response

Dear Editor,

We thank the Referee for making valuable comments and suggestions. In the revised manuscript we address the issues raised by the Referee by adding clarifications to the text in appropriate places and updating the figures and references.

The detailed response to the Referee’s comments and the corresponding revisions made in the manuscript are itemized below:

  1. In the first paragraph of the Introduction (page 2) we rephrase the sentence "Such a tiny interaction strength renders freeze-in dark matter invisible to the experiments" to "Such a tiny interaction strength makes freeze-in dark matter models challenging to detect at the experiments compared to the usual WIMP scenarios. For recent attempts to detect FIMP, see for example [30–33]." We have also added the relevant references including 1904.07915 and 1807.01730.

  2. We have included 1807.01730 in the list of references (Ref [31]) in contexts of detection of freeze-in scenarios (Introduction, page 2, end of 1st pargraph) and the dark photon portal models (Introduction, page 3, 2nd paragraph).

  3. In accordance with the Referee's suggestion we have updated Table 1 with an additional column detailing the origin of the dissipation rates with appropriate references.

  4. We have cited 2007.04328 in appropriate places and explicitly mention where we have reproduced their results (page 5, after Eq.2.4) and where we differ from their parametrizations (footnote 3, page 4).

  5. The discussion on page 7 (page 8 in the updated manuscript) is clarified as per the Referee's suggestion. In the updated manuscript we choose benchmark parameters such that the dark matter yield approximately saturates the observed relic abundance in the radiation dominated universe. In contrast the DM remains under-abundant in a matter dominated universe for the same set of parameters. We have introduced a new Eq.3.7 to estimate the dilution of the DM relic density in the EMD scenario. This clearly indicates that a larger coupling strength between the DM and the SM particles would be required to reproduce the correct freeze-in relic density for this case. We have modified Fig.1 and Fig.2 with new set of parameters to clarify this point.

  6. As commented by the Referee, we have indeed checked that additional processes do not spoil the existing freeze-in condition. In particluar, we ensure that the dark photon does not achieve thermal equilibrium via the inverse decay processes. As a consequence, number changing processes within the dark sector can be safely neglected. In the manuscript we elucidate this point in two different places for two scenarios, (i) M_A'> 2m_\chi (page 9, paragraph starting with "In this range of parameters...") and (ii) M_A'< 2m_\chi (page 10, last paragraph of section 4).

  7. We thank the Referee for pointing out the references on supernova bounds. We indeed found out that the constraints given in the 1st reference (1901.08596) are relevant for our parameter space. We have included it in the right panel of Fig.3. On the other hand we observe that the limits given in the 2nd reference (1905.09284) do not provide any additional constraint than the existing bound on the parameter space relevant for our paper.

We hope that the modified manuscript will be considered for publication.

Sincerely, Avik Banerjee (On behalf of all authors)

---

## Round 3 · List of Changes

Point by point list of changes in the revised manuscript:

  1. The last sentence of the first paragraph of the Introduction (page 2) is rephrased in accordance with the Refree's comment.

  2. We have included 1509.01598 (Ref[30]), 1807.01730(Ref [31]), and 1904.07915 (Ref[32]) in the list of references.

  3. As per the Referee's suggestion we have updated Table 1 (page 4) with an additional column detailing the origin of the dissipation rates with appropriate references.

  4. We have cited 2007.04328 in appropriate places and explicitly mention where we have reproduced their results (page 5, after Eq.2.4) and where we differ from their parametrizations (footnote 3, page 4).

  5. The discussion on page 8 is expanded with a new equation (Eq.3.7) for clarification as per the Referee's suggestion. In the same context, Fig.1 and Fig.2 are also modified with new set of parameters for the ease of discussion. The figure captions are updated accordingly.

  6. Text added in page 9, paragraph starting with "In this range of parameters..." and page 10, last paragraph of section 4 in context of the point 6 raised by the Referee.

  7. The right panel of Fig.3 is updated with a new constraint (purple region) given in 1901.08596 (Ref[106]), as suggested by the Referee.

---

## Editorial Decision

published